# Anti-Melanogenic Effect of Ethanolic Extract of *Sorghum bicolor* on IBMX–Induced Melanogenesis in B16/F10 Melanoma Cells

**DOI:** 10.3390/nu12030832

**Published:** 2020-03-20

**Authors:** Hye Ju Han, Seon Kyeong Park, Jin Yong Kang, Jong Min Kim, Seul Ki Yoo, Ho Jin Heo

**Affiliations:** Division of Applied Life Science (BK21 plus), Institute of Agriculture and Life Science Gyeongsang National University, Jinju 52828, Korea; gksgpwn2527@naver.com (H.J.H); tjsrud2510@naver.com (S.K.P.); kangjy2132@naver.com (J.Y.K.); myrock201@naver.com (J.M.K.); ysyk9412@naver.com (S.K.Y.)

**Keywords:** anti-melanogenesis, B16/F10 melanoma cell, hydroxyoctadecadienoic acid, Sorghum bicolor, 3-isobutyl-1-methylxanthine

## Abstract

To evaluate possibility as a skin whitening agent of *Sorghum bicolor* (*S. bicolor*), its antioxidant activity and anti-melanogenic effect on 3-isobutyl-1-methylxanthine (IBMX)-induced melanogenesis in B16/F10 melanoma cells were investigated. The result of total phenolic contents (TPC) indicated that 60% ethanol extract of *S. bicolor* (ESB) has the highest contents than other ethanol extracts. Antioxidant activity was evaluated using the 2,2’-azino-bis-(3-ethylbenzothiazolin-6-sulfonic acid) diammonium salt (ABTS)/1,1-diphenyl-2-picryl-hydrazyl (DPPH) radical scavenging activities and malondialdehyde (MDA) inhibitory effect. These results showed ESB has significant antioxidant activities. Inhibitory effect against tyrosinase was also assessed using L-tyrosine (IC_50_ value = 89.25 μg/mL) and 3,4-dihydroxy-L-phenylalanine (L-DOPA) as substrates. In addition, ESB treatment effectively inhibited melanin production in IBMX-induced B16/F10 melanoma cells. To confirm the mechanism on anti-melanogenic effect of ESB, we examined melanogenesis-related proteins. ESB downregulated melanogenesis by decreasing expression of microphthalmia-associated transcription factor (MITF), tyrosinase and tyrosinase-related protein (TRP)-1. Finally, 9-hydroxyoctadecadienoic acid (9-HODE), 1,3-*O*-dicaffeoylglycerol and tricin as the main compounds of ESB were analyzed using the ultra-performance liquid chromatography-ion mobility separation-quadrupole time of flight/tandem mass spectrometry (UPLC-IMS-QTOF/MS^2^). These findings suggest that ESB may have physiological potential to be used skin whitening material.

## 1. Introduction

Melanin is generated through a process, called as melanogenesis within the intracellular melanosomes of melanocytes, and the production and distribution of melanin determines the skin, eyes and hair color. In addition, melanin plays a vital role in protecting skin from various molecules and stimuli such as ultraviolet (UV) radiation, reactive oxygen species (ROS), α-melanocyte-stimulating hormone (α-MSH) and cyclic adenosine monophosphate (cAMP) elevating agents including forskolin and IBMX [1]. In particular, UV radiation directly and indirectly damages skin cells via ROS generation such as hydroxyl radicals, superoxide anions, and hydrogen peroxide [2]. The generated ROS causes single- and double-strand DNA breaks, and attacks important cellular structural molecules such as proteins and lipids [2]. Thus, melanin and antioxidant enzymes such as glutathione peroxidase, superoxide dismutase and catalase maintain a constant level by removing the ROS produced. However, excessive ROS production as a result of various factors activates melanogenesis in skin, and the abnormal melanin accumulation induces negative hyperpigmentation effects, causing age or liver spots, blotches and freckles [3,4]. Currently, natural compounds such as kojic acid, hydroquinone, arbutin and ascorbic acid, which are known as melanin synthesis inhibitors, are being utilized as skin whitening agent additives. However, these ingredients not only have poor penetration into skin, but also cause cytotoxicity and inflammation with long-term administration [5]. In this regard, it is necessary to develop a nonirritating and effective whitening agent to treat human skin hyperpigmentation, as well as to study natural resource materials for this.

Human skin cells commonly generate melanin as a photoprotection reaction. Melanogenesis is a complex process that is regulated via a variety of signaling pathways, and all signals ultimately upregulate MITF. As activated MITF promotes the tyrosinase gene family (tyrosinase and TRPs) expression, melanogenesis begins in earnest [6]. Generally, the first step in melanogenesis is that tyrosinase catalyzes the oxidation of L-tyrosine to 3,4-dihydroxy-L-phenylalanine (L-DOPA), and subsequently oxidizes L-DOPA into DOPA quinone. And then TRP-2 converts DOPA chrome to 5,6-dihydroxyindole-2-carboxylic acid (DHICA) or 5,6-dihydroxyindole (DHI). Finally, DHICA and DHI are oxidized by TRP-1, eventually leading to melanin production [6].

IBMX and other methylxanthines stimulate melanogenesis by inhibiting phosphodiesterase and thereby increasing levels of cAMP [7]. cAMP activated by IBMX phosphorylates extracellular-signal-regulated kinase (ERK) and phosphoinositide 3-kinases/protein kinase B (PI3K/Akt) signaling pathways, and ultimately leads to the activation of MITF in melanogenesis processes [7]. Therefore, the downregulation of these proteins can be regarded as an important factor for the anti-whitening effect.

*Sorghum bicolor* (*S. bicolor*), which belongs to the family Poaceae, is considered an important crop in tropical regions, including Africa, Central America, and arid regions, and is the fourth most important cereal crop of the world after wheat, rice and maize [8]. Recent studies have shown that *S. bicolor* is beneficial to humans because it contains phytochemicals such as phenolic compounds, tannins and anthocyanins [9]. These phytochemicals have gained increased attention due to its antioxidant, anti-obesity, anti-diabetes and anti-inflammatory effects [10,11,12,13,14]. While there are various research results, studies on the skin whitening effect of *S. bicolor* have not yet been conducted. Therefore, the present study aimed to investigate the antioxidant activity and anti-melanogenic effects of *S. bicolor*, and to examine their effects on IBMX-induced melanogenesis in B16/F10 melanoma cells.

## 2. Materials and Methods

### 2.1. Chemicals

Folin–Ciocalteu’s reagent, L-DOPA, L-tyrosine, L-ascorbic acid, arbutin, acarbose, dimethyl sulfoxide (DMSO), bovine serum albumin, α-glucosidase, mushroom tyrosinase, bovine serum albumin, 3-(4,5-dimethyl-2-thiazolyl)-2,5-diphenyltetrazolium bromide (MTT) and IBMX were purchased from Sigma-Aldrich Chemical Co. (St. Louis, MO, USA). MITF (bsm-51339M) and tyrosinase (bs-0819R) were purchased from Bioss (Beijing, China). TRP-1 (sc-166857) and β-actin (sc-69879) were purchased from Santa Cruz Biotechnology (Dallas, TX, USA). Secondary antibodies (anti-rabbit and anti-mouse) were obtained from Cell Signaling Technology (Danvers, MA, USA).

### 2.2. Sample Preparation

*S. bicolor* (20 g) was extracted with various ethanol concentration (0, 20, 40, 60, 80 and 95%; 1L) at 40℃ for 2 h using a reflux condenser. The extracts were filtered using a filter paper (Whatman International Limited, Kent, UK), and concentrated using a vacuum evaporator (N-1000; EYELA Co., Tokyo, Japan). After, the extracts were lyophilized using a vacuum freeze dryer (Il Shin Lab Co., Ltd., Yangju, Korea), and the dried extracts were stored at -20℃ until used.

### 2.3. In Vitro Antioxidant Activity

#### 2.3.1. Total Phenolic Contents (TPC)

Total phenolic contents were examined based on the principle that Folin–Ciocalteu’s reagent is reduced to blue reaction product under alkaline conditions. A sample (1 mL) mixed with Folin–Ciocalteu’s reagent and 7% sodium carbonate. The mixture was activated for 2 h, and then the absorbance was measured at 760 nm using a spectrophotometer (UV-1201; Shimadzu, Kyoto, Japan). TPC was calculated from the standard curve of gallic acid and the results were expressed as mg GAE g^−1^.

#### 2.3.2. Radical Scavenging Activity

ABTS radical cation solution was produced by mixing 2.45 mM potassium persulfate and 7 mM ABTS with 100 mM potassium phosphate buffer (pH 7.4) containing 150 mM and allowing them to react for 24 h at room temperature. The ABTS solution was then diluted with distilled water to obtain an absorbance of 0.700 ± 0.020 at 734 nm. The sample was allowed to react with 980 μL the ABTS solution for 10 min at 37℃ and then absorbance at 734 nm was measured using a spectrophotometer (UV-1201; Shimadzu, Kyoto, Japan).

DPPH radical solution was prepared by dissolving 0.1 mM DPPH in 80% methanol. The DPPH solution was diluted to an absorbance of 1.000 ± 0.020 at 517 nm. 50 μL of the sample was mixed with 1.45 mL of the DPPH solution and reacted for 30 min in the dark. After reacting, the mixture was determined at 517 nm.

#### 2.3.3. Inhibitory Effect on Lipid Peroxidation

To measure the inhibitory effect on lipid peroxidation in brains tissue, the thiobarbituric acid (TBA) reactive substance method was used. Brain tissue was homogenated in 20 mM Tris-HCl buffer (pH 7.4), and centrifuged at 6000× *g* for 20 min. The supernatant was added to 0.1 mM L-ascorbic acid and 10 μM ferrous sulfate 37℃ for 1 h incubation. Next, 30% trichloroacetic acid and 1% TBA were added to the mixture, which was then incubated in a water bath at 80℃ for 20 min. Then, the TBA-MDA complex was measured using a spectrophotometer (UV-1201; Shimadzu, Kyoto, Japan) at 532 nm.

### 2.4. Tyrosinase Inhibitory Effect

The tyrosinase inhibitory effect was determined using L-tyrosine as a substrate. A sample was added to a 96-well plate and mixed with 0.1 M sodium phosphate buffer, tyrosinase and 0.1 mM L-tyrosine substrate to react at 37℃. After incubating, enzyme activity was measured using a microplate reader (EPOCH2; BioTek, Winooski, VT, USA) at 490 nm. Also, L-DOPA was used as a substrate to measure tyrosinase inhibitory activity. 67 mM sodium phosphate buffer, tyrosinase and 10 mM L-DOPA substrate were added to the sample to react at 37℃ for 10 min. Tyrosinase activity was measured at 415 nm.

### 2.5. α-Glucosidase Inhibitory Effect

The α-glucosidase inhibitory effect was measured by mixing 0.1 M sodium phosphate buffer and α-glucosidase at 37℃for 10 min. After activating, the mixture was added to 5 mM 4-nitrophenyl-α-D-glucopyranoside in buffer at 37℃ for 5 min, and then the absorbance was measured at 405 nm using a microplate reader (EPOCH2; BioTek, Winooski, VT, USA).

### 2.6. Cell Viability Assay

The cytotoxicity of the sample was estimated using an MTT assay. B16/F10 melanoma cells were cultured at 1×10^4^ cells/well in a 96-well plate for 24 h. Thereafter, samples were treated to 1, 2, 5 and 10 μg/mL concentrations. Samples were treated for 48 h, after which the cells were treated with 10 μg/mL of MTT stock solution for 3 h. Finally, media was removed, and DMSO was added to solubilize the formazan. The amount of formazan was quantified using a microplate reader (EPOCH2; BioTek, Winooski, VT, USA) at 570 nm (excitation wavelength) and 655 nm (emission wavelength).

### 2.7. Measurement of Cellular Melanin Contents

To measure the amount of melanin, B16/F10 melanoma cells were cultured in a 24-well plate at a density of 4×10^4^ cells/well above for 24 h. Then, the cells were treated with 100 mM IBMX and ESB or positive control (arbutin) for an additional 48 h. After treatment, the cells were washed and harvested using phosphate buffer saline (PBS). The harvested cells were then centrifuged at 16,000× *g* for 10 min, and the obtained pellet was dissolved with 1 N sodium hydroxide containing 10% DMSO at 90 ℃ for 20 min. The melanin contents were measured at 490 nm using a microplate reader (EPOCH2; BioTek, Winooski, VT, USA).

### 2.8. Western Blot Analysis

Pretreated B16/F10 melanoma cells were washed with PBS and lysed using RIPA buffer containing 1% protease inhibitors on ice. Protein concentrations were determined by Bradford protein assay, and the equal proteins were denatured with a Laemmli buffer for 10 min at 95℃. The denatured proteins were separated by 10% sodium dodecyl sulfate polyacrylamide gel electrophoresis (SDS-PAGE), and then were transferred onto a polyvinylidene difluoride (PVDF) membranes (Millipore, Billerica, MA, USA). Next, membranes were blocked with 5% skim milk in Tris-buffered saline with 0.1% Tween 20 (TBST) for 1 h, and then reacted with primary antibodies overnight at 4℃. The membranes were reacted with the secondary antibodies for 1 h and the immune complexes were visualized using an enhanced chemiluminescence reagent (Bionote, Hwaseong, Korea). The band images were detected and analyzed by Chemi-doc (iBright Imager, Thermo-Fisher Scientific, Waltham, MA, USA).

### 2.9. UPLC-IMS-QTOF/MS^2^ Analysis

The main compounds in the 60% ethanolic extracts of *S. bicolor* were analyzed by UPLC-IMS-QTOF/MS^2^ (Vion, Waters Corp., Milford, MA, USA). Chromatographic separation was carried out using an Acquity UPLC BEH C18 column (2.1 × 100 mm, 1.7 μm particle size; Waters Corp.) at a flow rate of 0.35 mL/min. The mobile phases consisted of solvent A (0.1% formic acid in distilled water) and B (acetonitrile), and analysis conditions included the following time gradient: 1% B at 0–1 min, 1–100% B at 1–7 min, 100% B at 7–8 min, 100–1% B at 8–8.2 min, 1% B at 8.2–10 min. Mass spectrometry was performed in the electrospray ionization (ESI) negative mode under the following conditions: source temperature, 100 °C; desolvation line temperature, 400 °C; ramp collision energy, 10–30 V; capillary voltage, 2.5 kV. Phenolic compounds were detected by a full scan ranging from *m/z* 50 to 1500.

### 2.10. Statistical Analysis

All results were presented as the means ± standard deviation (SD). The statistically difference between groups were determined by A one-way analysis of variance (ANOVA) using Duncan’s new multiple-range test (*p* < 0.05) with SAS software version 9.4 (SAS Institute, Cary, NC, USA). And the interrelationship between the antioxidant effects (ABTS/DPPH radical scavenging activity and MDA inhibitory effect) and the inhibitory effect of melanogenesis-mediated enzymes (tyrosinase and α-glucosidase) was evaluated Pearson correlation analysis.

## 3. Results

### 3.1. Total Phenolic Contents

To compare antioxidant activity of each ethanolic extract from *S. bicolor*, TPC was investigated. As shown in Figure 1, TPC were different among the ethanolic extracts; the 60% ethanolic extract (150.08 ± 1.13 mg GAE/g) was significantly higher than the other extractions (0%; 70.83 ± 1.18, 20%; 100.42 ± 0.72, 40%; 114.67 ± 1.46, 80%; 118.33 ± 3.17 and 95%; 73.67 ± 1.46 mg GAE/g). Therefore, the 60% ethanolic extract was used for further experimental analyses.

### 3.2. Radical Scavenging Activity

The antioxidant activity of the 60% ethanolic extract from *S. bicolor* (ESB) was measured using the ABTS/DPPH assay and MDA assay, and the results were shown in Figure 2. The ABTS radical scavenging activity of ESB was exhibited 97.98% at 1000 μg/mL concentration, and vitamin C (99.82%), which is used as the positive control, and also had similar radical scavenging activity at same concentration (Figure 2A). And, ESB showed a 50% inhibition effects (IC_50_ value) of the ABTS radical at 409.71 μg/mL concentration. As shown in Figure 2B, the DPPH radical scavenging activity of ESB was showed 79.44% at 1000 μg/mL concentration, and the IC_50_ value was indicated at 612.53 μg/mL. The inhibitory effect of MDA production of ESB showed an inhibitory effect of 89.54% at 50 μg/mL concentration, and showed a higher inhibitory effect than the catechin (71.03%), which is the positive control, at the same concentration (Figure 2C). In addition, IC_50_ value of MDA was observed at 16.56 μg/mL concentrations of ESB.

### 3.3. Inhibitory Effect of Melanogenesis-Mediated Enzymes

The inhibitory effect of melanogenesis-mediated enzymes was evaluated using tyrosinase and α-glucosidase (Figure 3). To measure inhibitory activity of tyrosinase against melanin synthesis, experiments were performed using L-tyrosine and L-DOPA as a substrate under cell-free conditions. As a result, the IC_50_ value was 89.25 μg/mL when L-tyrosine was used as a substrate (Figure 3A), but L-DOPA could not that more than 50% of inhibitory effect was observed (Figure 3B). At this time, the IC_50_ value of the positive control (arbutin) for L-tyrosine was 74.35 μg/mL, which was higher than ESB. ESB was found to have a better inhibitory effect on tyrosinase using L-tyrosine than using L-DOPA as a substrate. Therefore, based on the results of assay using two substrates, the tyrosinase inhibitory effect of ESB is considered to be related to the oxidation of L-tyrosine to L-DOPA rather than the oxidation of L-DOPA to DOPA quinone in melanogenesis. Tyrosinase inhibitory effect using L-DOPA as substrate and radical scavenging activity (ABTS; 0.943, *p* < 0.05, DPPH; 0.952, *p* < 0.05) indicated the positive strong correlations in Table 1.

To measure the inhibitory effect of melanogenesis-mediated enzyme, the inhibitory activity against α-glucosidase of ESB was measured. As shown in Figure 3C, ESB inhibited α-glucosidase by 33.72% at 200 μg/mL concentration. And the IC_50_ of ESB was 46.29 μg/mL, which is about five times higher than acarbose (216.05 μg/mL). These results indicate that ESB has better α-glucosidase inhibitory activity than acarbose as a positive control. And α-glucosidase inhibitory effect revealed strong positive correlations with MDA inhibitory effect (0.966, *p* < 0.05) in Table 1.

### 3.4. Cell Viability and Cellular Melanin Synthesis

To confirm the cytotoxicity of ESB, B16/F10 melanoma cells were treated with ESB for 24 h at 1, 2, 5, 10 and 20 μg/mL concentrations. The results showed that ESB did not affect cell viability at the indicated concentrations when compared to the control (Figure 4A). Therefore, experiments were carried out by selecting 2, 5 and 10 μg/mL concentrations. Subsequently, B16/F10 melanoma cells were treated with IBMX to analyze the effect of ESB on melanin production. As shown in Figure 4B, the level of melanin content increased to 316.85% when treated with IBMX. On the other hand, the ESB treated group (108.60%) at 10 μg/mL concentration showed significantly decreased melanin contents compared to the IBMX treated group, and similar melanin contents compared to the arbutin group (101.79%) as a positive control.

### 3.5. Melanogenesis Pathway in B16/F10 Melanoma Cells

Melanogenesis is an essential process to protect human skin from external stimuli, and this process is regulated by various mechanisms. In this study, we measured the expression level of melanogenesis-regulating molecules to confirm the potential inhibitory mechanism of ESB on IBMX-induced melanin production in B16/F10 melanoma cells (Figure 5). The results showed that ESB effectively decreased IBMX-induced MITF expression (Figure 5A,B). Thus, the expression of tyrosinase (Figure 5A,C) and TRP-1 (Figure 5A,D) was downregulated by reduced MITF with ESB treatment. As a result, ESB treatment downregulated the expression of related proteins in IBMX-induced melanogenesis in B16/F10 melanoma cells.

### 3.6. UPLC-IMS-QTOF/MS^2^ Analysis

A profile of the main compounds of ESB was scanned by LC/MS in a range of *m/z* 50–1500, and the base peak chromatograms are shown in Figure 6. Further identification and characterization of the compounds was performed in comparison with literature data using MS^2^ fragmentation data (Table 2). Based on the fragments in previous literature, peak 2 was 1-*O*-caffeoylglycerol (*m/z* 253.07, 179.03, 161.02 and 135.04) [15]; peak 3, dicaffeoylglycerides (*m/z* 415.10, 253.07, 179.03 and 135.04) [16]; peak 4, 1,3-*O*-dicaffeoylglycerol (*m/z* 415.10, 253.07, 179.03, 161.02 and 135.04) [16]; peak 5, *p*-coumaroyl-caffeoylglycerol (*m/z* 399.10, 253.07, 179.03, 161.02 and 135.04) [17]; peak 6, feruloyl-caffeoylglycerol (*m/z* 429.12, 253.07, 193.05, 161.02 and 134.03) [18]; peak 7, tricin (*m/z* 329.23, 314.04, 299.01 and 271.02) [19]; and peak 9, 9-hydroxyoctadecadienoic acid (9-HODE) (*m/z* 295.22, 277.21 and 171.10) [20], respectively. Among them, 1,3-*O*-dicaffeoylglycerol (Figure 6B), tricin (Figure 6C) and 9-HODE (Figure 6D) were identified as major compounds.

## 4. Discussion

Melanogenesis is reported to be caused by increased oxidative stress from external stimuli. Oxidative stress causes oxidation of DNA and proteins, lipid peroxidation, and increased unsaturated fatty acids. This stress also leads to increase unnecessarily melanin synthesis in melanocytes on the skin. Thus, excessive melanin generated can cause pigmentation, and it can lead to skin cancer. IBMX stimulate melanogenesis by inhibiting phosphodiesterase and thereby increasing levels of cAMP [7]. As the level of cAMP in cells increases, it causes activation of the ERK and PI3K/Akt signaling pathways that promote the generation of proteins associated with melanogenesis. Therefore, we investigated the whitening effect of ESB on IBMX-induced melanogenesis in B16/F10 melanoma cells.

Phenolic compounds as the secondary metabolites of plants have the property of being stabilized by a resonance structure when they react with radicals [21]. Also, phenolic compounds act as important factors for the antioxidant activity such as ABTS/DPPH radical scavenging activity, and act as inhibitors of pigment formation because they have a chemical structure like that of L-tyrosine [22,23]. An ABTS/DPPH assay is the most common and simplest method for estimating in vitro antioxidant activity. In this study, ESB shown a high TPC (Figure 1) and ABTS/DPPH radical scavenging activity (Figure 2A,B). Especially, ESB was higher in the ABTS radicals scavenging activity than the DPPH radical scavenging activity. The DPPH assay is used to measure the radical scavenging activity of hydrophobic compounds, whereas the ABTS assay makes it possible to measure the radical scavenging activity against hydrophobic as well as hydrophilic compounds [24]. Therefore, it was confirmed that the radical scavenging activity of ESB was more affected by the hydrophilic compound.

UV radiation exposure causes ROS production and has direct and indirect effects on the skin. In particular, ROS induces lipid peroxidation by attacking unsaturated fatty acids, which are components of cell membranes [2]. The accumulation of lipid peroxides leads to the destruction of the antioxidant system of skin, causing pigmentation, inflammation and aging [25]. In this study, ESB showed an excellent inhibit lipid peroxidation (Figure 2C). The phenolic compounds contained in natural plants were reported to be associated with radical scavenging activity [26]. Also, Maresca et al. [27] reported that removal of ROS with antioxidants is effective in inhibiting melanin biosynthesis. A recent study reported that the ethyl acetate and aglycone fractions of *Dendropanax morbifera* leaf can act as skin cell protectants and natural antioxidants by protecting HaCaT cells against oxidative stress [28]. Moreover, treatment with the same fractions showed a better anti-melanogenic effect than arbutin, used as a positive control, against α-MSH-induced excessive melanin generation in B16/F10 melanoma cells [28].

Several mechanisms are associated with melanogenesis, and the common goal for skin whitening agents is the regulation of tyrosinase, which is an important enzyme in melanogenesis. Tyrosinase, a copper-containing enzyme, catalyzes two reactions in this pathway: the hydroxylation of L-tyrosine to L-DOPA and the oxidation of L-DOPA to DOPA quinone [6]. When exposed to large amounts of UV or ROS, the activity of tyrosinase is increased and melanin synthesis is also increased. Therefore, the results of measuring the inhibitory effect of tyrosinase in relation to melanin synthesis showed that ESB was found to have a better inhibitory effect on tyrosinase using L-tyrosine than using L-DOPA as a substrate (Figure 3A,B). A recent study reported the correlation between TPC and tyrosinase inhibition because phenolic compounds have a chemical structure similar to that of L-tyrosine, a substrate of tyrosinase. According to Choi et al. [29], it was observed that the efficiency of free radical scavenging activity and tyrosinase inhibitory effect increased in proportion to the fermentation time of *Cheonggukjang*, which was due to the amount of TPC increasing with fermentation time. Im & Lee [30] observed that the ethyl acetate fraction of 75% ethanolic extract from *Taraxacum coreanum* had abundant TPC and antioxidant activity, and the tyrosinase inhibitory effect was superior to other fractions (hexane, chloroform, butanol and aqueous).

Tyrosinase, one of the glycoproteins, is glycosylated when polypeptide chains translocate into the endoplasmic reticulum. Various enzymes are involved in this process, and among them, inhibition of α-glucosidase could inhibit the folding of tyrosinase to form an inactive structure without copper. As a result, tyrosinase is unable to be transported to the melanosome and inhibit melanin production. The polyphenol present in plants is capable not only of reducing oxidative stress, but also inhibits carbohydrate hydrolyzing enzymes such as α-glucosidase by binding to proteins [31]. A previous study indicated that B16/F10 melanoma cells in the presence of glycosylation inhibitors results in the reduction of melanogenesis by decreasing the total content of tyrosinase [32]. And Choi et al. [33] reported that deoxynojirimycin, one of the α-glucosidase inhibitors, blocked glycosylation of tyrosinase and inhibited tyrosinase migration to melanosomes. Ando et al. [34] have shown that it is possible to inhibit the melanin synthesis in melanoma cells by controlling the glycosylation of tyrosinase by α-glucosidase inhibitors such glutathione, ferritin and feldamycin. Kim et al. [35] reported the good correlations for Bee Pollen as natural antioxidant between TPC and tyrosinase inhibitory effect (0.973, *p* < 0.05), and also TPC gave the positive correlation with DPPH radical scavenging activity (0.897, *p* < 0.05). And the results suggest that polyphenols as natural antioxidants could be potential effective skin whitening based on their excellent antioxidant activity. In our study, antioxidant effect of ESB in the process of inhibiting the action of tyrosinase was indicated to be highly correlated through inhibition the oxidation of L-DOPA to DOPA quinone and the glycosylation of tyrosinase (Table 1). For this reason, the antioxidant effect of ESB has an effect on skin whitening effect by affecting the inhibition of melanogenesis-mediated enzyme.

The PKA signaling pathway is the main process involved in melanogenesis, and this is activated by IBMX, which is a cAMP elevating agent. The PKA signaling pathway can regulate transcription factors such as MITF and cAMP-responsive element-binding protein, which are involved in the expression of melanogenesis regulators such as tyrosinase, TRP-1 and TRP-2. Consequently, IBMX treatment increases the expression of MITF and the tyrosinase gene family (tyrosinase and TRP-1) through cAMP activation [7]. Park et al. [36] showed that the amount of melanin synthesis from *Papenfusiella kuromo* ethanolic extract (65.17 ± 13.40%) at a concentration of 40 μg/mL was similar to kojic acid (72.30 ± 3.92%) as a positive control, and reported that it showed potential as a natural whitening material. In this study, ESB showed an effective inhibitory effect on melanin synthesis similar to that of the positive control at low concentrations (Figure 4). After that, to assesses detailed anti-melanogenic effect of ESB, western blot analysis was performed on melanogenesis-mediated protein using B16/F10 melanoma cell.

Akt, which is involved in various mechanisms, plays a key role in multiple cellular processes such as apoptosis, glucose and skin metabolism. In particular, the increase of cAMP by IBMX treatment decreases Akt phosphorylation and its activation. Also, inactivated Akt induces the activation of GSK-3β, which means that phosphorylation is not achieved, that is, the level of *p*-GSK-3β is reduced [7]. According to previous literature, the aerial part of *Pueraria thunbergiana* downgrades MITF by activating the Akt/GSK-3β signaling pathway in B16/F10 melanoma cells [37]. Also, Huang et al. [38] showed that [6]-shogaol promoted *p*-Akt expression and inhibited melanogenesis signaling in B16/F10 melanoma cells. Activated GSK-3β upregulates the tyrosinase gene family by phosphorylating Ser289 of MITF, consequently promoting melanogenesis. MITF is a major transcription factor that binds to a tyrosinase gene promoter, and it subsequently increases the expression of enzymes related to melanocyte proliferation and melanogenesis [7]. When the expression of tyrosinase is promoted by MITF, it subsequently leads to an increase in the levels of TRP-1 and TRP-2, and as a result, these melanogenic enzymes produce melanin [6]. Oxidative stress caused by UV light stimulates melanin pigmentation in the skin. Therefore, the scavenging activity of radicals or ROS is effective to suppress melanin production, and polyphenols with antioxidant activity may have excellent whitening effects [38]. Choi et al. [39] showed that Korean bamboo stems (*Phyllostachys nigra* variety henosis) downregulated PKA/CREB-mediated MITF via scavenging of ABTS/DPPH radicals. And a 70% ethanolic extract of *Nelumbo nucifera* G. Leaf inhibited the expression of MITF, tyrosinase and TRPs with antioxidant activities such as electron donation ability and inhibition of xanthine oxidase [40]. In this experiment, it was shown that ESB treatment decreased the expression of MITF and tyrosinase gene family (tyrosinase and TRP-1), which promote the oxidation of DHICA to DHI in the melanogenesis pathway (Figure 5). Based on these results, ESB has been shown to downregulate IBMX-mediated melanogenesis based on its superior antioxidant effects.

Based on the previous literature, the main phenolic compounds in ESB were identified that 1,3-*O*-dicaffeoylglycerol (Figure 6B), tricin (Figure 6C) and 9-HODE (Figure 6D) [16,19,20]. Among them, 9-HODE, which is the most abundant compound in ESB, is one of the metabolites of linoleic acid, and shows a whitening effect on irritated skin by inhibiting tyrosinase [41]. Also, unlike other aliphatic compounds, 9-HODE is reported to be stable in all oxidation experiments and to have a whitening effect on ROS damage [41]. Meanwhile, tricin (5,7-dihydroxy-2-(4-hydroxy-3,5-di-methoxyphenyl)-4H-chromen-4-one) has long been recognized for its beneficial health effects, such as antioxidant, antiviral, and antitumor/anticancer activities [42,43,44]. Mu et al. [45] reported that tricin exhibited a better inhibitory effect on tyrosinase activity compared with arbutin as a positive control. Moreover, Meng et al. [46] reported that tricin isolated from the methanol extract of young green barley (*Hordeum vulgare* L.) inhibited melanin biosynthesis and tyrosinase activity in B16/F10 melanoma cells through the hydroxyl group at the C-4’ position and methoxy groups at the C-3’,5’ positions of the tricin skeleton. Phenylpropanoids such as 1-*O*-caffeoylglycerol, dicaffeoylglycerides, and 1,3-*O*-dicaffeoylglycerol are organic compounds that are synthesized by plants from phenylalanine and tyrosine. Phenylpropanoid and their glycosylated forms were reported to be potent antioxidants either by directly scavenging ROS, or by acting as chain-breaking peroxyl radical scavengers [47].

*p*-Coumaric acid, which has a chemical structure similar to L-tyrosine, competes with L-tyrosine for active sites on tyrosinase [48]. It is known as a strong tyrosinase inhibitor, and it was reported that its anti-melanogenesis effect is stronger than structurally similar compounds such as cinnamic acid [48]. In addition, An et al. [48] reported that *p*-coumaric acid, a constituent of *Sasa quelpaertensis* Nakai, inhibited tyrosinase activity using L-DOPA as a substrate, and reduced melanin production in B16/F10 melanoma cells. Therefore, ESB may be considered to have an excellent whitening effect through *p*-coumaric acid. Furthermore, the MS spectra of 1-*O*-caffeoylglycerol and 1-3-*O*-dicaffeoyl glycerol showed similar fragmentation patterns at m/z 179, 161 and 135, and these fragments were reported as caffeic acid residue. They have a UV spectrum similar to that of a caffeic acid derivative with a λ_max_ at about 326 nm, and so are considered to be a caffeic acid derivative [48]. It was reported that caffeic acid, a derivative of various compounds, is a potent antioxidant *in vitro*/*in vivo*, and reduces tyrosinase activity and melanogenesis in B16/F10 melanoma cells. Consequently, the anti-melanogenic effect of ESB might be considered to be due to antioxidants and skin whitening substances such as aliphatic and phenolic compounds.

## 5. Conclusions

To evaluate possibility as a skin whitening agent of *Sorghum bicolor*, antioxidant, and anti-melanogenic effects of ESB on IBMX-induced melanogenesis were evaluated in B16/F10 melanoma cells. ESB exhibited significant antioxidant activities in ABTS/DPPH radical scavenging activity and MDA inhibitory effect. ESB prevented the first step of melanogenesis by inhibiting α-glucosidase and tyrosinase using L-tyrosine and L-DOPA as substrates. The anti-melanogenic effects of ESB on IBMX-induced melanogenesis were evaluated in B16/F10 melanoma cells, and the results showed that ESB inhibited IBMX-induced melanogenesis in B16/F10 melanoma cells. Melanin accumulation was inhibited by downregulation of the expression of MITF and the tyrosinase gene family (tyrosinase and TRP-1) in IBMX-induced melanogenesis. The major compounds of ESB were analyzed as 1,3-*O*-dicaffeoylglycerol, tricin and 9-HODE. Consequently, ESB showed *in vitro* antioxidant activity and an anti-melanogenic effect in B16/F10 melanoma cells, and thus, could be used as a skin whitening agent in the cosmetics market.

## Figures and Tables

**Figure 1 nutrients-12-00832-f001:**
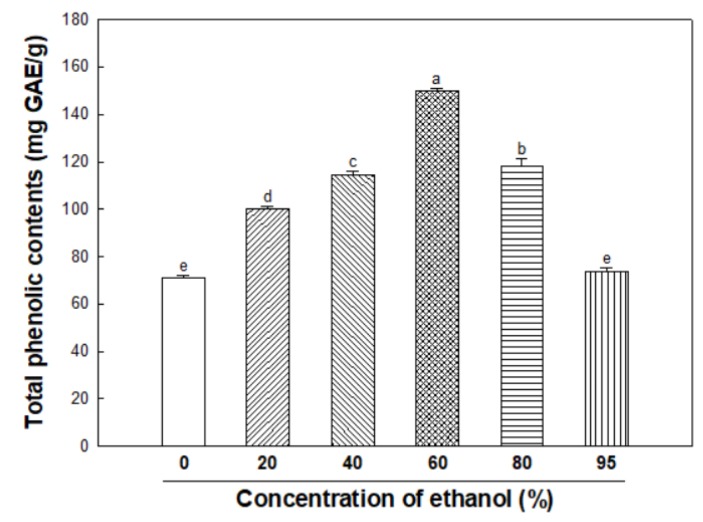
Total phenolic contents of various ethanolic extracts from *Sorghum bicolor*. Data were expressed as the means ± SD (*n* = 3). Each small letter represented statistical difference, and was statistically considered at *p* < 0.05.

**Figure 2 nutrients-12-00832-f002:**
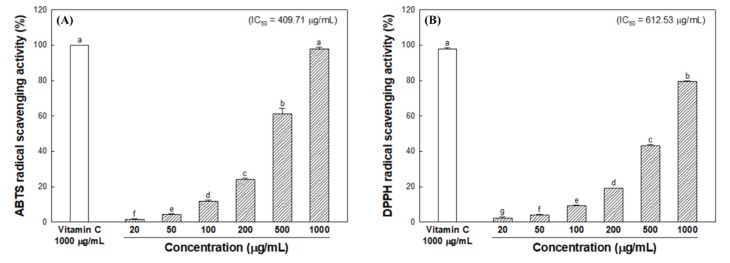
2,2’-azino-bis-(3-ethylbenzothiazolin-6-sulfonic acid) diammonium salt (ABTS) radical scavenging activity (**A**), 1,1-diphenyl-2-picryl-hydrazyl (DPPH) radical scavenging activity (**B**) and malondialdehyde (MDA) inhibitory effect (**C**) of the 60% ethanolic extract from *Sorghum bicolor* (ESB). Data were expressed as the means ± SD (*n* = 3). Each small letter represented statistical difference, and was statistically considered at *p* < 0.05.

**Figure 3 nutrients-12-00832-f003:**
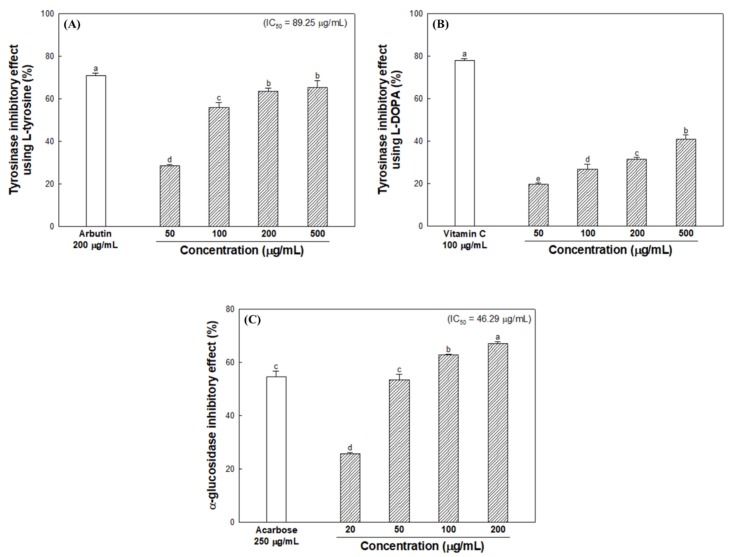
Inhibitory effect of the 60% ethanolic extract of *Sorghum bicolor* (ESB) on melanogenesis-mediated enzymes. Tyrosinase inhibitory effect using L-tyrosine (**A**) and L-DOPA (**B**) as substrates, and α-glucosidase inhibitory effect (C). Data were expressed as the means ± SD (*n* = 5). Each small letter represented statistical difference, and was statistically considered at *p* < 0.05.

**Figure 4 nutrients-12-00832-f004:**
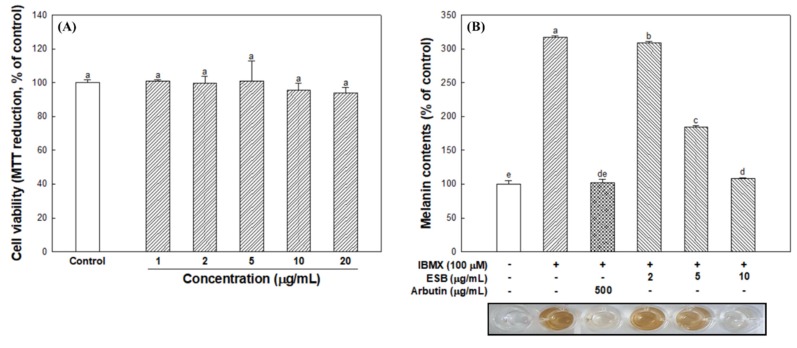
Cell viability (**A**) and melanin contents (**B**) in B16/F10 melanoma cells of the 60% ethanolic extract of *Sorghum bicolor* (ESB). Data were expressed as the means ± SD (*n* = 3). Each small letter represented statistical difference, and was statistically considered at *p* < 0.05.

**Figure 5 nutrients-12-00832-f005:**
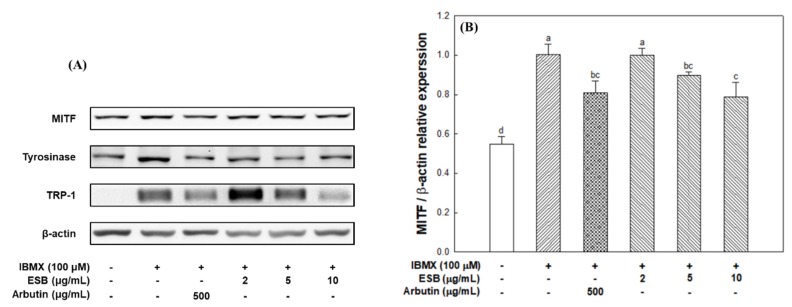
Anti-melanogenic effect of the 60% ethanolic extract of *Sorghum bicolor* (ESB) on 3-isobutyl-1-methylxanthine (IBMX)-induced melanogenesis in B16/F10 melanoma cells. Representative band images of proteins (**A**), relative expression of microphthalmia-associated transcription factor (MITF) (**B**), tyrosinase (**C**) and tyrosinase and tyrosinase-related protein (TRP)-1 (**D**). Data were expressed as the means ± SD (*n* = 3). Each small letter represented statistical difference, and was statistically considered at *p* < 0.05.

**Figure 6 nutrients-12-00832-f006:**
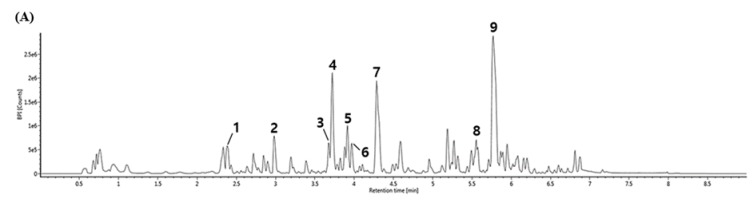
UPLC IMS-QTOF/MS spectra in negative ion mode (**A**) and MS^2^ spectra and chemical structure of 1,3-*O*-dicaffeoylglycerol (**B**), Tricin (**C**) and 9-hydroxyoctadecadienoic acid (9-HODE) (**D**) from the 60% ethanolic extract of *Sorghum bicolor* (ESB).

**Table 1 nutrients-12-00832-t001:** Correlation between antioxidant activity and inhibitory effect of melanogenesis-mediated enzyme

	Pearson Correlation		
	Tyrosinase Inhibitory Effect		α-Glucosidase Inhibitory Effect
	L-tyrosine	L-DOPA	
**ABTS radical scavenging activity**	0.900	0.943^*^	0.792
**DPPH radical scavenging activity**	0.718	0.952^*^	0.937
**MDA inhibitory effect**	0.988	0.995	0.966^*^

*Correlation is significant (*p* < 0.05).

**Table 2 nutrients-12-00832-t002:** Identification of main compounds in the 60% ethanolic extract of *Sorghum bicolor* (ESB).

No.	RT (min)	[M-H]^-^ (m/z)	MS^2^ Fragments (m/z)	Proposed Compound
1	2.38	625.19	407.11, 383.11, 221.06, 125.02, 89.02	Unknown
2	2.98	253.07	179.03, 161.02, 135.04	1-*O*-caffeoylglycerol
3	3.68	415.10	253.07, 179.03, 135.04	Dicaffeoylglycerides
4	3.72	415.10	253.07, 179.03, 161.02, 135.04	1,3-*O*-dicaffeoylglycerol
5	3.91	399.10	253.07, 179.03, 161.02, 135.04	*p*-coumaroyl-caffeoylglycerol
6	3.97	429.12	253.07, 193.05, 161.02, 134.03	Feruloyl-caffeoylglycerol
7	4.29	329.23	314.04, 299.01, 271.02	Tricin
8	5.55	315.25	297.24, 279.23	Unknown
9	5.77	295.22	277.21, 171.10	9-hydroxyoctadecadienoic acid (9-HODE)

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
