# Peer review of "Anti-Melanogenic Effect of Ethanolic Extract of Sorghum bicolor on IBMX–Induced Melanogenesis in B16/F10 Melanoma Cells"

_nutrients, 2020, doi:10.3390/nu12030832_

Round 1
Reviewer 1 Report
Author presented results on the antioxidant capacity of Sorghum bicolor, inhibiting melanin generation, and component analysis of Sorghum bicolor extracts in this manuscript, however, there is no correlation between each results. Especially, I don't know why author didantioxidant tests and UPLC-IMS-QTOF/MS2 to check the whitening effect.
Author Response
Thank your very much for the valuable comments. Please see the attachment.
[The replies on the review of ‘# Reviewer 1’]
Author presented results on the antioxidant capacity of Sorghum bicolor, inhibiting melanin generation, and component analysis of Sorghum bicolor extracts in this manuscript, however, there is no correlation between each result. Especially, I don't know why author did antioxidant tests and UPLC-IMS-QTOF/MS2 to check the whitening effect.
→ Thanks for your valuable opinion. There seems to be insufficient explanation for correlation between antioxidant effect and whitening effect. Therefore, we carefully checked and analyzed the ‘Pearson correlation’ between the antioxidant effects (ABTS/DPPH radical scavenging activity and MDA inhibitory effect) and the inhibitory effect of melanogenesis-mediated enzymes (tyrosinase and α-glucosidase). And also, the analyzed results were added in ‘results and discussion’ session.
[Materials and Methods; Lines 174-180]
2.10. Statistical analysis
All results were presented as the means ± standard deviation (SD). The statistically difference between groups were determined by A one-way analysis of variance (ANOVA) using Duncan's new multiple-range test (p<0.05) with SAS software version 9.4 (SAS Institute, Cary, NC, USA). And the interrelationship between the antioxidant effects (ABTS/DPPH radical scavenging activity and MDA inhibitory effect) and the inhibitory effect of melanogenesis-mediated enzymes (tyrosinase and α-glucosidase) was evaluated Pearson correlation analysis.
[Results; Lines 220-221]
Tyrosinase inhibitory effect using L-DOPA as substrate and radical scavenging activity (ABTS; 0.943, p<0.05, DPPH; 0.952, p<0.05) indicated the positive strong correlations in Table 1.
[Results; Lines 226-235]
And α-glucosidase inhibitory effect revealed strong positive correlations with MDA inhibitory effect (0.966, p<0.05) in Table 1.
Table 1. Correlation between antioxidant activity and inhibitory effect of melanogenesis-mediated enzyme
|
|
Pearson correlation |
||
|
|
Tyrosinase inhibitory effect |
α-glucosidase inhibitory effect |
|
|
|
L-tyrosine |
L-DOPA |
|
|
ABTS radical scavenging activity |
0.900 |
0.943* |
0.792 |
|
DPPH radical scavenging activity |
0.718 |
0.952* |
0.937 |
|
MDA inhibitory effect |
0.988 |
0.995 |
0.966* |
*Correlation is significant (p < 0.05).
[Discussion; Lines 355-363]
Kim et al. [35] reported the good correlations for Bee Pollen as natural antioxidant between TPC and tyrosinase inhibitory effect (0.973, p<0.05), and also TPC gave the positive correlation with DPPH radical scavenging activity (0.897, p<0.05). And the results suggest that polyphenols as natural antioxidants could be potential effective skin whitening based on their excellent antioxidant activity. In our study, antioxidant effect of ESB in the process of inhibiting the action of tyrosinase was indicated to be highly correlated through inhibition the oxidation of L-DOPA to DOPA quinone and the glycosylation of tyrosinase (Table 1). For this reason, the antioxidant effect of ESB has an effect on skin whitening effect by affecting the inhibition of melanogenesis-mediated enzyme.
[Reference]
- Kim, S.B.; Jo, Y.H.; Liu, Q.; Ahn, J.H.; Hong, I.P.; Han, S.M.; Hwan, B.Y.; Lee, M.K. Optimization of extraction condition of bee pollen using response surface methodology: correlation between anti-melanogenesis, antioxidant activity, and phenolic content. Molecules 2015, 20, 19764-19774. https://doi.org/10.3390/molecules201119656
→ Thanks for your valuable opinion on the UPLC-IMS-QTOF/MS2 analysis to check the whitening effect. We have speculated that the whitening effect based on antioxidant effect, enzymatic inhibitory activity and anti-melanogenic activity of ESB may be due to phenolic compounds as physiological material of Sorghum bicolor in ‘discussion’ section. Therefore, we tried to identify the major physiological material including ESB using UPLC-IMS-QTOF/MS2 system. And then, we tried to verify the whitening effects of the identified materials using previous scientific literature.
Thanks for your precious time for reviewing our manuscript!
Reviewer 2 Report
The manuscript is interesting but I'm not sure whether the subject of manuscript match to the aim and scope of Nutrients. The investigated plant is indeed used as food product but the activity investigated by Authors is associated with the external application of Sorghum bicolor extract on skin. Moreover, the experimental section should be strongly completed. The section about plant material and extraction procedure should be added.
Minor comments:
Line 180 – 182 is not clear.
The quality of figures should be improved and fonts should be bigger because they are hardly to follow.
Figure 3 is cited in wrong place.
The concentration ranges for experiments are different. Why? The ranges presented in 3.3. section are too narrow. IC 50 is not calculated (Fig 3b).
Line 262 - 265 the phrases “ ESB containing the physiological compounds against oxidative stress” and “Phenolic compounds, (…), contain flavonoids, phenolic acids and phytoestrogens.” are not clear and should be reedited.
The discussion section are too long and should be given in more compact form e.g. the sentence in line 274-276 is unnecessary because this information was given in line 270-272.
Line 277 the phrase “…..phenol combined with these..” is not clear.
What identified compound is responsible for activity of extract?
Author Response
Thank you very much for the valuable comments. Please see the attachment.
[The replies on the review of ‘# Reviewer 2’]
The manuscript is interesting but I'm not sure whether the subject of manuscript match to the aim and scope of Nutrients. The investigated plant is indeed used as food product but the activity investigated by Authors is associated with the external application of Sorghum bicolor extract on skin.
→ Thanks for your valuable comments. The aim and scope of journal is very important factor for publication, so we carefully thought about that of ‘Nutrients’. We thought that edible foods are included to safe substances for human body. If we assessed the functionality of their nutrients (e.g. phenolic compound, vitamins etc.), we thought it could also help the skin function through oral-intake as functional foods. In this regard, we believed that our manuscript could be matched the aim and scope of ‘Nutrients’ by providing the valuable information as the skin whitening materials of Sorghum bicolor through functional evaluation.
Moreover, the experimental section should be strongly completed. The section about plant material and extraction procedure should be added.
→ Thanks for your valuable comments. We carefully checked and added the plant material and extraction information in ‘material and methods’ section.
[Materials and Methods; Lines 85-90]
2.2. Sample preparation
- bicolor (20 g) was extracted with various ethanol concentration (0, 20, 40, 60, 80 and 95%; 1L) at 40℃ for 2 h using a reflux condenser. The extracts were filtered using a filter paper (Whatman International Limited, Kent, UK), and concentrated using a vacuum evaporator (N-1000; EYELA Co., Tokyo, Japan). After, the extracts were lyophilized using a vacuum freeze dryer (Il Shin Lab Co., Ltd., Yangju, Korea), and the dried extracts were stored at -20℃ until used.
[Minor comments]
1) Line 180 – 182 is not clear.
[Lines 180-182 → Lines 199-200]
As shown in Figure 2B, the result of DPPH radical scavenging activity was investigated 79.44% at 1000 μg/mL concentration, and showed the IC50 value at 612.53 μg/mL concentration.
→ As shown in Figure 2B, the DPPH radical scavenging activity of ESB was showed 79.44% at 1000 μg/mL concentration, and the IC50 value was indicated at 612.53 μg/mL.
2) The quality of figures should be improved and fonts should be bigger because they are hardly to follow.
→ We have carefully checked your suggestion, and then have increased the font size of all the Figures to make the data more visible.
Figure 1
Figure 2
Figure 3
Figure 4
Figure 5
3) Figure 3 is cited in wrong place.
[Lines 190-193 → Lines 209-213]
To measure inhibitory activity against the oxidation associated with melanin synthesis, experiments were performed using L-tyrosine and L-DOPA as a substrate under cell-free conditions (Figure 3).
→ 3.3. Inhibitory effect of melanogenesis-mediated enzymes
The inhibitory effect of melanogenesis-mediated enzymes was evaluated using tyrosinase and α-glucosidase (Figure 3). To measure inhibitory activity of tyrosinase against melanin synthesis, experiments were performed using L-tyrosine and L-DOPA as a substrate under cell-free conditions.
4) The concentration ranges for experiments are different. Why? The ranges presented in 3.3. section is too narrow.
→ We have carefully checked your valuable suggestion. We conducted various concentration ranges for each experiment. And optimal concentrations are shown in the figure according to each result based on the concentration that indicate 50% inhibition.
- IC50 is not calculated (Fig 3b).
In Fig. 3B, we conducted the tyrosinase assay using L-DOPA as a substrate at concentrations of 500 μg/mL or higher, but had difficulty identifying more than 50% of the inhibitory effect. For this reason, we could not calculate the IC50 value.
5) Line 262 - 265 the phrases “ESB containing the physiological compounds against oxidative stress” and “Phenolic compounds, (…), contain flavonoids, phenolic acids and phytoestrogens.” are not clear and should be reedited.
→ According to your valuable suggestion, we carefully checked and reedited the phrases.
[Lines 262-263 → Lines 287-290]
Therefore, in this study, we investigated the anti-melanogenic effect of ESB containing the physiological compounds against oxidative stress and melanogenesis using IBMX.
→ Therefore, we investigated the whitening effect of ESB on IBMX-induced melanogenesis in B16/F10 melanoma cells.
[Lines 264-265 → Lines 291-294]
Phenolic compounds, which are the secondary metabolites of plants, contain flavonoids, phenolic acids and phytoestrogens. And they have the property of being stabilized by a resonance structure when they react with radicals [21].
→ Phenolic compounds as the secondary metabolites of plants have the property of being stabilized by a resonance structure when they react with radicals [21].
6) The discussion section is too long and should be given in more compact form e.g. the sentence in line 274-276 is unnecessary because this information was given in line 270-272.
→ According to your valuable suggestion, we carefully checked and revised by deleting unnecessary sentences in ‘discussion’ section.
[Lines 268-279 → Lines 297-310]
An ABTS/DPPH assay is the most common and simplest method for estimating in vitro antioxidant activity. The DPPH assay is used to measure the radical scavenging activity of hydrophobic compounds, whereas the ABTS assay makes it possible to measure the radical scavenging activity against hydrophobic as well as hydrophilic compounds [24]. In this study, ESB shown a high TPC (Figure 1) and ABTS/DPPH radical scavenging activity (Figure 2A,2B). Especially, ESB was higher in the ABTS radicals scavenging activity than the DPPH radical scavenging activity. This is because the ABTS assay has the advantage of measuring both the activity of hydrophilic and hydrophobic compounds [24]. In addition, Jo et al. [25] reported that radical scavenging ability may differ depending on the type of phenol combined with these, due to the radical difference between ABTS (cation radical) and DPPH (free radical). Therefore, it was confirmed that the radical scavenging activity of ESB was more affected by the hydrophilic compound.
→ An ABTS/DPPH assay is the most common and simplest method for estimating in vitro antioxidant activity. In this study, ESB shown a high TPC (Figure 1) and ABTS/DPPH radical scavenging activity (Figure 2A,2B). Especially, ESB was higher in the ABTS radicals scavenging activity than the DPPH radical scavenging activity. The DPPH assay is used to measure the radical scavenging activity of hydrophobic compounds, whereas the ABTS assay makes it possible to measure the radical scavenging activity against hydrophobic as well as hydrophilic compounds [24]. Therefore, it was confirmed that the radical scavenging activity of ESB was more affected by the hydrophilic compound.
[Lines 289-292 → Lines 320-324]
Moreover, treatment with the same fractions showed a better anti-melanogenic effect than arbutin, used as a positive control, against α-MSH-induced excessive melanin generation in B16/F10 melanoma cells [28]. Based on these results suggest that ESB has excellent antioxidant activity, and protection of skin against oxidative stress and whitening effects can be expected.
[Lines 318-323 → Lines 350-355]
Ando et al. [34] has been shown to inhibit the melanin synthesis in melanoma cells by controlling the glycosylation of tyrosinase by α-glucosidase inhibitors such glutathione, ferritin and feldamycin. In this study, ESB has shown better α-glucosidase inhibitory effect than acarbose as a positive control (Figure 4). Based on these results, ESB has been shown to inhibit melanogenesis-mediated enzymes, such as tyrosinase and α-glucosidase, based on its superior antioxidant effects.
[Lines 324-342 → Lines 364-380]
The PKA signaling pathway is the main process involved in melanogenesis, and this is activated by IBMX, which is a cAMP elevating agent. The PKA signaling pathway can regulate transcription factors such as MITF and cAMP-responsive element binding protein, which are involved in the expression of melanogenesis regulators such as tyrosinase, TRP-1 and TRP-2. Consequently, IBMX treatment increases the expression of MITF and the tyrosinase gene family (tyrosinase and TRP-1) through cAMP activation [7]. Melanin plays an important role in determining skin color, but excessive accumulation can cause problems such as pigmentation. Therefore, it is necessary to develop whitening agents that inhibit melanin production by external stimuli. On this point, the most commonly used functional cosmetics materials are kojic acid, hydroquinone, arbutin and ascorbic acid. However, prolonged use of these substances on the skin is reported to cause toxicity such as skin penetration and stability [5]. In order to find a substitute for commercialized functional cosmetic materials, research is currently being conducted to find natural substances with potential whitening effects. Park et al. [36] showed that the amount of melanin synthesis from Papenfusiella kuromo ethanolic extract (65.17 ± 13.40%) at a concentration of 40 μg/mL was similar to kojic acid (72.30 ± 3.92%) as a positive control, and reported that it showed potential as a natural whitening material. In this study, ESB showed an effective inhibitory effect on melanin synthesis similar to that of the positive control at low concentrations (Figure 54). After that, to assesses detailed anti-melanogenic effect of ESB, western blot analysis was performed on melanogenesis-mediated protein using B16/F10 melanoma cell.
7) Line 277 the phrase “…..phenol combined with these..” is not clear.
→ We have carefully checked your valuable suggestion. However, we thought that this sentence was not only unclear, but also repeated with previous sentence. For this reason, we have removed the duplicate sentence.
[Lines 269-279 → Lines 297-310]
An ABTS/DPPH assay is the most common and simplest method for estimating in vitro antioxidant activity. The DPPH assay is used to measure the radical scavenging activity of hydrophobic compounds, whereas the ABTS assay makes it possible to measure the radical scavenging activity against hydrophobic as well as hydrophilic compounds [24]. In this study, ESB shown a high TPC (Figure 1) and ABTS/DPPH radical scavenging activity (Figure 2A,2B). Especially, ESB was higher in the ABTS radicals scavenging activity than the DPPH radical scavenging activity. This is because the ABTS assay has the advantage of measuring both the activity of hydrophilic and hydrophobic compounds [24]. In addition, Jo et al. [25] reported that radical scavenging ability may differ depending on the type of phenol combined with these, due to the radical difference between ABTS (cation radical) and DPPH (free radical). Therefore, it was confirmed that the radical scavenging activity of ESB was more affected by the hydrophilic compound.
→ An ABTS/DPPH assay is the most common and simplest method for estimating in vitro antioxidant activity. In this study, ESB shown a high TPC (Figure 1) and ABTS/DPPH radical scavenging activity (Figure 2A,2B). Especially, ESB was higher in the ABTS radicals scavenging activity than the DPPH radical scavenging activity. The DPPH assay is used to measure the radical scavenging activity of hydrophobic compounds, whereas the ABTS assay makes it possible to measure the radical scavenging activity against hydrophobic as well as hydrophilic compounds [24]. Therefore, it was confirmed that the radical scavenging activity of ESB was more affected by the hydrophilic compound.
8) What identified compound is responsible for activity of extract?
→ Thanks for your valuable opinion. There is very important to find out which compound is responsible for the whitening activity of the extract. In our results, major compounds of Sorghum bicolor were identified as 1-O-caffeoylglycerol, dicaffeoylglycerides, 1,3-O-dicaffeoylglycerol, p-coumaroyl-caffeoylglycerol, feruloyl-caffeoylglycerol, tricin and 9-HODE, as shown in '3.6. UPLC-IMS-QTOF/MS2 analysis' section. To determine which compound from the extract have a whitening effect, each single compound will be prepared by prep-HPLC etc. However, we could not purchase or isolate all major compounds. In near future, we are going to isolate and get main compounds from ESB using preparative liquid chromatography (Prep/LC). After that, we will evaluate their activity to assess whether the effects of ESB are due to major compounds or minor compounds or their combined actions.
Thanks for your precious time for reviewing our manuscript!
Round 2
Reviewer 1 Report
Previous review was not fully answered.
Because it is unclear why ingredients is identified using mass analysis and whether the extracts regulates oxidation or expression melanogenesis gene expression.
Author Response
Thanks again for your precious time for reviewing our manuscript!
We carefully checked on your valuable opinion. And our opinion was mentioned below.
As we mentioned in ‘Introduction and Discussion’, melanin is generated through a process, called as melanogenesis within the intracellular melanosomes of melanocytes, and the production and distribution of melanin determines the skin, eyes and hair color. Melanin plays a vital role in protecting skin from various molecules and stimuli such as ultraviolet (UV) radiation and reactive oxygen species (ROS) etc. In addition, UV radiation directly and indirectly damages skin cells via ROS generation such as hydroxyl radicals, superoxide anions, and hydrogen peroxide. In particular, excessive ROS production as a result of various factors activates melanogenesis in skin, and the abnormal melanin accumulation induces hyperpigmentation effects. Recent studies have shown that Sorghum bicolor (S. bicolor) is beneficial to humans because it contains phytochemicals such as phenolic compounds, tannins and anthocyanins etc. Among of them, phenolic compounds as the secondary metabolites of plants have the property of being stabilized by a resonance structure when they react with radicals. In this point, the antioxidant effect of S. bicolor was investigated, and analyzed its major compounds (Fig. 6 & Table 2) to check the relation between antioxidant function and molecules of the extracts.
As another mechanism, melanogenesis-regulating molecules (proteins) were measured to confirm the potential inhibitory mechanism of the extracts on IBMX-induced melanin production in B16/F10 cells (Fig. 5).
However, unfortunately, we did not purify or prepare major compounds reported to have antioxidant effects. Therefore, the correlation between antioxidant effect and gene (protein) expression for melanogenesis could not be confirmed.
Consequently, in view of these findings, we just described as follows in the ‘Conclusion’.
‘To evaluate possibility as a skin whitening agent of Sorghum bicolor, antioxidant and anti-melanogenic effects of ESB on IBMX-induced melanogenesis were evaluated in B16/F10 melanoma cells.’
‘ESB showed in vitro antioxidant activity and an anti-melanogenic effect in B16/F10 melanoma cells, and thus could be used as a skin whitening agent in the cosmetics market.’
To determine which compound from the extract have a whitening effect, each single compound will be prepared by prep-HPLC etc. However, we could not purchase or isolate all major compounds. In near future, we are going to isolate and get main compounds from ESB using preparative liquid chromatography (Prep/LC). After that, we will evaluate the relation between antioxidants and main proteins for melanogenesis.
Additionally, our manuscript was revised by a native reviewer (http://www.supreme-trans.co.kr), and the certificate was attached.
Reviewer 2 Report
I accept all responses
Author Response
Thanks again for your precious time for reviewing our manuscript!
Additionally, our manuscript was revised by a native reviewer (http://www.supreme-trans.co.kr), and the certificate was attached.